# Targeting Dietary and Microbial Tryptophan-Indole Metabolism as Therapeutic Approaches to Colon Cancer

**DOI:** 10.3390/nu13041189

**Published:** 2021-04-03

**Authors:** Madhur Wyatt, K. Leigh Greathouse

**Affiliations:** 1Human Health, Performance and Recreation, Robbins College of Health and Human Sciences, Baylor University, Waco, TX 76798-7346, USA; madhur_wyatt1@baylor.edu; 2Human Science and Design, Robbins College of Health and Human Sciences, Baylor University, Waco, TX 76798-7346, USA

**Keywords:** tryptophan metabolism, kynurenine, indoleamine 2,3-dioxygenase, aryl hydrocarbon receptor, microbiome, indole, colon cancer

## Abstract

Tryptophan metabolism, via the kynurenine (Kyn) pathway, and microbial transformation of tryptophan to indolic compounds are fundamental for host health; both of which are altered in colon carcinogenesis. Alterations in tryptophan metabolism begin early in colon carcinogenesis as an adaptive mechanism for the tumor to escape immune surveillance and metastasize. The microbial community is a key part of the tumor microenvironment and influences cancer initiation, promotion and treatment response. A growing awareness of the impact of the microbiome on tryptophan (Trp) metabolism in the context of carcinogenesis has prompted this review. We first compare the different metabolic pathways of Trp under normal cellular physiology to colon carcinogenesis, in both the host cells and the microbiome. Second, we review how the microbiome, specifically indoles, influence host tryptophan pathways under normal and oncogenic metabolism. We conclude by proposing several dietary, microbial and drug therapeutic modalities that can be utilized in combination to abrogate tumorigenesis.

## 1. Introduction

Over a century ago (1901), English chemist Frederick Gowland Hopkins made a significant contribution to the field of nutrition by discovering the essential amino acid Tryptophan (Trp) from casein, a milk protein. As an essential amino acid, Trp can only be obtained through diet, mainly meat, dairy, and seeds. Trp is required for protein and niacin biosynthesis and is a precursor of serotonin and melatonin. Apart from these fundamental functions, Trp is appreciated for its influence on both host and microbial metabolism via two distinct pathways: the Kynurenine (Kyn) Pathway and Indolic Pathway. These pathways metabolize Trp into metabolic and neuroactive compounds that influence microbial composition and host physiology in an enzyme-dependent manner. Indole has a bicyclic ring formed by benzene and pyrrole groups. This indole structure is part of Trp and makes it unique in its structure and function. Indole can be kept intact or cleaved from the Trp structure, generating bioactive molecules [1]. Specific taxa among the microbiota also take advantage of this unique property of Trp to produce indoles for signaling and defense [2]. Microbial and dietary indoles can also trigger or suppress immune function thereby maintaining a symbiotic relationship with the host [3,4]. However, both Kyn and indolic processes can become dysfunctional in the pathogenesis of colon cancer.

In this review, we begin by describing the different metabolic pathways of Trp and their impact on the host physiology. We then discuss the alterations that occur in these pathways during colon cancer. Next, we introduce the significance of the microbial degradation of Trp in healthy conditions, along with the repercussions of microbial alterations in carcinogenesis. We conclude by highlighting the gaps in knowledge regarding microbial-host Trp metabolism and propose dietary, microbial and drug targets that could have therapeutic applications.

## 2. The Four Main Pathways of Human Trp Metabolism

There are four known metabolic pathways that Trp can enter once in the upper gastrointestinal (GI) tract; Kyn pathway, serotonin pathway, indolic pathway (bacterial degradation) and tryptamine pathway [5]. These distinct pathways compete for the available free Trp pool and specifically convert Trp into Kyn, serotonin, indoles, and tryptamine as the end products, respectively [6]. About 90–95% [7,8] of Trp is converted into Kyn and downstream metabolites (Figure 1) via the Kyn pathway. These metabolites include L-kynurine acid, 3-hydroxykynurine, anthralinic acid, quinolinic acid, 3-hydroxyanthralininc acid, picolinic acid, most of which produce NAD+ and ATP in host cells, and play a vital role in inflammation, immune tolerance, and neurotransmission [5,9,10,11].

About 4–6% of unabsorbed Trp then moves along the gastrointestinal tract and is metabolized by the microbiome into indole and indolic compounds including indole-3 pyruvate, indole-3-acetamide, indole-3-acetaldehyde, indole-3-acetic acid, indole-3-lactic acid, and indole-3- propionic acid [8,12,13,14,15,16]. These indolic compounds can bind to pregnane X receptors (PXR) and aryl hydrocarbon receptor (AhR) to promote intestinal homeostasis, enhance barrier function and tight junctions, reduce permeability, regulate intestinal immune tolerance [5,13,17,18]. Additionally, indoles can act as antioxidants and neuroprotective compounds [19,20]. Overall, indole production has an important impact on host health as a result of Trp metabolism by the microbiota [5]. 

In addition to Kyn pathway and microbial degradation, 1–2% of Trp can be converted into 5-hydroxytryptamine (serotonin) in the enterochromaffin cells of the intestinal mucosa. Serotonin is primarily involved in regulating gut motility, vasodilation, and maintenance of mood [21]. A minor contribution of Trp (about 1%) is towards the production of tryptamine, which can be initiated by both the host and microbiota [22]. Microbial tryptamine, a ligand for serotonin receptors, can also affect gut motility and transit time, given the abundance of such receptors in the GI tract [23]. Contribution of the gut microbiota towards serotonin and tryptamine production is an understudied field of research and will need further investigation in the context of carcinogenesis.

## 3. The Kynurenine Pathway Is the Major Tryptophan Degradation Pathway

The Kyn pathway (90–95% of ingested Trp) is the major Trp degradation pathway that produces Kyn and other neuroactive metabolites either in the liver or in the extrahepatic tissue. In addition to neuroactive metabolites, the Kyn pathway also regulates systemic Trp levels, availability of Trp for serotonin synthesis, vitamin B3, and hepatic heme synthesis [9]. About 90% of Trp metabolized via the Kyn pathway takes place in the liver where all the enzymes for complete transformation of Trp into NAD+ are present. The hepatic Kyn pathway is mainly induced by signaling from glucocorticoids, as well as estrogen and glucagon [5,24,25]. The remaining 10% of the Trp transformation takes place via the Kyn pathway in the extrahepatic tissues, and is induced by cytokines (Interferon-γ (IFN-γ), Interleukin-1 (IL-1), Interluekin-6 (IL-6)), lipopolysaccharides, prostaglandins and amyloid peptides [7]. However, the metabolites produced in the extrahepatic tissues do not contain all the enzymes for complete conversion of Trp into NAD+. Availability, or lack thereof, of all enzymes to completely transform Trp determines the intermediates that are produced, ultimately affecting the functional outcome [9,26,27,28]. 

The array of metabolites produced via the Kyn pathway have led many researchers to investigate Trp and the role of these metabolically active metabolites in immune function, behavioral disorders including mood, anxiety and depression, inflammatory conditions, gut homeostasis, inflammatory bowel disease (IBD), neurodegenerative disorders such as Alzheimer’s disease, and cancer. All these disease states appear to have an altered or amplified Kyn pathway activity [1,10,29,30,31,32,33,34]. Under inflammatory conditions, common to most of these disease states, Trp is shunted into the Kyn pathway, especially in the extrahepatic tissue (mostly immune cells), while the hepatic pathway activity is reduced [9,25,35]. This shunting of Trp into Kyn pathway, reduces Trp availability for other pathways especially serotonin and indolic pathways; the two pathways which have shown to be beneficial for host health in the presence of Trp [34,36,37]. Furthermore, reduced Trp and increased Kyn, alters the activation and balance of innate and adaptive immune cells towards a tolerogenic milieu [37]. Therefore, Kyn pathway activity is amplified under inflammatory conditions, in part due to an NAD+ requirement for increased energy demands in the immune cells and to implement immunomodulatory functions [35].

### 3.1. Kynurenine Pathway and the First-Rate Limiting Enzyme IDO1

The first and rate-limiting enzymes in Trp catabolism are Tryptophan 2,3-dioxygenase (TDO) and Indoleamine 2,3-dioxygenase (IDO1), which are present in the liver and in the extrahepatic tissues, respectively. TDO regulates systemic levels of Trp. Under normal physiological conditions, the majority of Trp is metabolized in the liver to produce Kyn, C -reactive proteins, haptoglobin, and fibrinogen [25]. IDO1 is expressed by various different tissue and cells types including lungs, placenta, GI tract, immune cells (e.g., monocytes, macrophages, dendritic cells, antigen-presenting cells) and epithelial cells. The function of IDO1 in extrahepatic cells is the same, which is the conversion of Trp into Kyn [38]. 

Under normal conditions, IDO1 expression regulates T cell proliferation to prevent tissue damage and reduce oxidative stress [25]. During inflammation, the expression of TDO is reduced and IDO1 expression is increased. Once IDO1 is activated upon signaling from cytokines such as IFN-γ, TNFα, prostaglandins and lipopolysaccrides, it converts tryptophan into N-formylkynurenine, followed by a rapid transformation into Kyn, the first stable catabolite in the pathway [5,39,40]. Kyn is then acted upon by various enzymes in a tissue-dependent manner and produces downstream neuroactive and immunoactive metabolites which regulate immune cell activity [41]. 

Extrahepatic Kyn production governs immune homeostasis [42]. This action occurs by reduction of activated T-cells, dendritic cells and, natural killer cells and induction of Th1 cell apoptosis to control excessive inflammation [11,40]. Each downstream Kyn metabolites perform specific functions. Kynurenic acid elicits an anti-inflammatory response through its anti-oxidant properties, while picolinic acid exhibits anti-tumor activity by suppressing T-cell and c-Myc activation [11,43,44,45]. Furthermore, 3-hydroxyanthralinic acid and quinolinic acid can act as neurotoxins in certain disease states including chronic brain injury, osteoporosis, coronary heart disease, Huntington’s disease, stroke, depression, and colon cancer [40,46,47]. Collectively, these downstream metabolites modulate immune homeostasis in part by activating the ligand-dependent transcription factor aryl hydrocarbon receptor (AhR). 

### 3.2. Kynurenine Pathway and AhR Activation

AhR is a transcription factor that belongs to the family of basic helix-loop-helix transcription factors that control genes containing xenobiotic response elements (XREs), as well as through non-XRE response elements; estrogen receptor and retinoic acid receptors [48]. In an inactive state, AhR is bound to chaperone proteins. Upon ligand binding, the chaperone protein disassociates with the complex and cytoplasmic AhR translocate into the nucleus heterodimerizing with AhR nuclear translocator (ARNT), leading to transcription of CYP1A1, CYP1B1, IDO1, TDO, IL-22, GSTA and AhRR (Table 1) [49,50]. Several of these genes control metabolism of xenobiotic and environmental chemicals including dioxins, polycyclic aromatic hydrocarbons, benzathracens and halogenated aromatic hydrocarbons through regulation of cytochrome P-450 [51,52,53,54,55]. Furthermore, AhR can interact with additional transcription factors, including pro-inflammatory nuclear factor-kB, and epigenetic regulators, as well as, acting as an E3 ubiquitin ligase. In the intestinal epithelial cells, AhR works to control immune homeostasis, epithelial barrier function, and symbiosis with the microbiota [48]. 

AhR activation is a crucial mediator for intestinal immunity and intestinal homeostasis through the development and regulation of intraepithelial lymphocytes and innate lymphoid cells [13,56,57,58,59,60]. AhR activation can be induced by distinct classes of endogenous ligands including Trp metabolites, dietary compounds, microbial Trp metabolites, bilirubin, arachidonic acid, prostaglandins, and cytokines (Table 1). Depending upon the ligand, AhR activation can result in different molecular and physiological responses including histone modification, lipid synthesis, energy, xenobiotic metabolism, immune function, epithelial barrier function and cell migration [56,57,58,59,60,61]. Therefore, AhR is involved in multiple pathways that regulate endogenous and exogeneous signals that controls the immune response. 

Several reports have shown the importance of AhR ligands in modulating gastrointestinal homeostasis via mucosal immune cells. However, the exact mechanism of this modulation is unclear due to the multitude of different ligands and transcriptional factors involved in controlling physiological effects [50]. AhR-modulated intestinal homeostasis can be achieved in one of two ways 1) it can exert an anti-inflammatory response through the activation of Tregs; 2) it can enhance intestinal mucosal integrity through the activation of Th17 cells and induction of IL-22 [13,50,62,63]. Therefore, AhR activation is critical in maintaining intestinal homeostasis. Thus, disruption in AhR expression or activity can lead to altered intestinal homeostasis and carcinogenesis. 

## 4. Contribution of Host Trp Metabolism in Colon Carcinogenesis

The alterations in Trp metabolism begin early on in colon carcinogenesis and permits immune evasion in tumor microenvironment (Figure 2). Recently, the oncogene c-MYC was shown to contriubte to this immune evasion through alteration of Trp metabolism. Specifically, c-MYC accelerated Trp uptake by upregulating Trp transporters SLC7A5 and SLC1A5 in colon cancer cells and tissues [43]. The study also showed an increase in the cytoplasmic levels of IDO1, arylformamidase (hydrolyzes N-formyl-L-Kyn to L-Kyn), and Kyn, which drive T cell inactivation and increase protein synthesis, in pathogenesis of colorectal cancer (CRC). Other studies have also demonstrated increased levels of Kyn in colon cancer patients when compared to healthy individuals [11]. Interestingly, the levels of enzymes involved in further degradation of Kyn into downstream metabolites were not elevated, suggesting Kyn to be the predominant metabolite elevated in CRC [43]. Our data (unpublished) also indicate an alteration in Trp metabolism in the microbiome of individuals with CRC compared to healthy individuals, which creates additional complexity in the relationship with the host Kyn pathway. Thus, understanding the contribution of the microbial Trp-Kyn pathway to colon carcinogenesis will be important for future drug discovery and treatment.

Excessive Kyn levels in the tumor microenvironment is one of the adaptive mechanism for the tumors to escape immune surveillance and metastasize via T-cell inactivation (Figure 3) [11,64,65]. Kyn drives tumor escape by acting as a ligand and constitutively activating AhR to express genes involved in cell-growth. Collectively, the alterations in Trp transporters and degrading enzymes leads to an increased Kyn pool within the tumor microenvironment. The excessive Kyn levels aid in tumorigenesis in two ways; 1), a portion of the Kyn produced can directly induce T-cell inactivation and apoptosis leading to immune evasion; 2) the remaining Kyn can constitutively activate AhR, transcribing genes for not only tumor escape but also for proliferation and metastasis [26,43,53,66,67]. Thus, the Kyn-AhR signaling pathway is one of the major contributors to colon cancer [33,34,36,38,40,43,57].

Over stimulation of the Kyn pathway can also be induced by stress and inflammation [68]. Recent evidence indicates that pro-inflammatory cytokines, including IFN-γ, can induce IDO1 gene expression due to presence of IFN response element. Increased IDO1 expression, especially in immune cells can enhance the activity of Tregs and suppress effector T-cells leading to immune tolerance [69]. Most individuals with cancer, including hepatocellular carcinoma, colon cancer, kidney cancer lung cancer among other, often exhibit increased Kyn levels and decreased serum Trp levels, or increased kyn/Trp ratio, which in part is explained by excessive inflammatory signaling molecules [11,70,71,72]. Evidence of this increased kyn/Trp ratio was recently shown in a study of individuals with colon cancer. Fecal metabolites from these patients demonstrated both an increased kyn/Trp ration, but also a decreased indole/Trp ratio (Venkateswaran & Conacci-Sorrell, 2020), suggesting loss of either indole producing microbes and/or an increase in microbes producing kynurinene metabolites. Indeed, *Pseudomonas aeruginosa* produces kynurenic acid and 3-OH-Kyn, which may disrupt the Trp metabolism of the host, and thus the immune response [73]. Additionally, administration with PEGylated kynureninase, a therapeutic enzyme, has shown to inhibit tumor growth by degradation of Kyn into non-toxic metabolites [74]. Therefore, it is becoming widely known, but not completely understood, that alterations in Trp metabolism is one of the key players in overexpression of IDO1 and Kyn. This alteration further leads to constitutive activation of AhR, perpetuating cell-growth. However, what is less well known is how the gut microbiota and their metabolites contribute to IDO1 expression, AhR activation, and immune cell balance. Though recently, the short chain fatty acid butyrate, produced by several commensal bacteria, was shown to decrease expression of IDO1 through reduction of STAT1 and its histone deacetylases inibitory activity, indicating that bacterial metabolites are able to control Trp metabolism [75].

The versatility of AhR binding has increased the research interest in its ability to regulate the immune system, intestinal homeostasis, and cancer progression. Under normal conditions, AhR activation results in transcription of genes, CYP1A1, CYP1B1 and AhRR (Aryl-Hydrocarbon Receptor Repressor), that control AhR activity in negative feedback loop. However, dysregulation of this feedback loop allows uncontrolled AhR activation, which is correlated with inflammatory processes including IBD, multiple sclerosis, cardiovascular conditions, and allergic responses, and carcinogenesis [76]. The multifaceted role of AhR in carcinogenesis is complex and only beginning to be understood. This, in part, is due to the ability of AhR to bind to many different ligand types, resulting in either enhancing or suppressing carcinogenesis [11,77]. Loss of AhR has been shown to enhance colon carcinogenesis, while re-establishing normal AhR activation inhibits the formation of polyps [48,62]. Reduced AhR expression has been seen in individuals with IBD, colitis, Crohn’s disease and CRC [78]. Alterations in Trp metabolism in part explain the reduced AhR availability; a hallmark of inflammatory disease states including CRC. Indeed, Crotti et al. demonstrated that alterations in Trp metabolism occur early in the pathogenesis of colon cancer preceding malignancy [36]. Specifically, it is understood that colon epithelial cells undergo changes in Trp metabolism within the tumor microenvironment as an adaptive mechanism to escape immune surveillance. While under normal physiology the AhR ligands exhibit a protective role by either inducing or by suppressing inflammation, in the context of carcinogenesis, loss of AhR or constitutive activation abrogates this protective function. Therefore, the metabolic end products of the Kyn pathway control the inflammatory process, in part through activation of AhR [36,79,80,81,82,83,84]. However, the mechanism that leads to these alterations in Trp metabolism and loss or dysfunctional AhR activity in colon carcinogenesis is not well understood.

## 5. Contribution of the Microbiota in Trp Metabolism

The gut microbiome forms a symbiotic relationship which affects host physiology in several ways. These effects include maintaining gut homeostasis, activating, or suppressing immune function, metabolic and neurological homeostasis. Microbial communities promote host health not only by producing vitamins, degrading bile, and oxalates, but also by synthesizing anti-inflammatory metabolites. These bacterial metabolites can act as neurotransmitters, signaling and secretory molecules, immunoregulators, neurotoxic and antimicrobial molecules directly affecting host physiology. One such metabolite group are indoles, first discovered and isolated by a German scientist Adolf Von Baeyers in 1866 in an indigo reduction process [85]. Indoles are produced through bacterial degradation of dietary Trp by the action of bacterial enzyme tryptophanase (TnaA) or degradation of phenylalanine by phenyllacetate dehydratase (fldAIBC), though other undiscovered pathways may be involved [86,87,88,89]. Over 85 bacterial species express TnaA, with some Peptostreptococcus and Clostridium species containing fldAIBC gene clusters [2,85,88]. Multiple types of indole derivatives can be produced from Trp due to expression of species-specific enzymes. The most prominent bacterial indoles include indole-3-acetamide, indole-3-acetaldehyde, indole-3-pyruvate, indole-3-aldehyde, indole-3-aceticacid, tryptamine, indole-3-propionic acid, and indole acrylic acid. Indole and indole derivatives play an important signaling role between the host and the microbiota and are a critical component of bacterial function. Within the microbial communities in the gut, indole production affects spore formation, plasmid stability, biofilm formation, antibiotic tolerance, cell division, and virulence [13]. Within the host, indoles activate signaling pathways that result in changes in intestinal epithelial barrier function, reduce permeability, promote immune tolerance, evict pathogens, reduce inflammation, and, control mucin production [12,85,90,91,92,93,94,95]. Colonization of germ-free mice for 4 weeks with *C. sporogenes* or a mutant (fldC) lacking the an enzyme necessary for indole (indole-3-propionic acid (IPA)) production, resulted in serum concentrations of 80 μM in mice colonized with wild-type but undetectable levels in mice colonized with the mutant strain. Further, colonization with the fldC mutant increased gut permeability, and lead to higher levels of circulating neutrophils, T cells (CD4+ and CD8+) and sIgA (lumen) [86]. Together, these results indicate that microbiota are capable of producing biologically relevant levels of indoles, and inducing changes in innate and adaptive immune response that may be important in colon cancer pathogenesis.

Indolic compounds can also act as AhR ligands to promote both pro- and anti-inflammatory effects [17,96]. Indoles can reduce expression of pro-inflammatory signals, IL-8 and NF-κB, and promote expression of anti-inflammatory cytokines including IL-10 [95]. In addition, indolic compounds regulate intestinal homeostasis through induction of IL-22, which improves barrier function, however, in the context of cancer (later stages) IL-22 production can promote tumor progression [97,98]. Microbial indole metabolites are normally a consistent source of host AhR ligands, unlike dietary sources that are transient [99]. The majority of indoles and indole derivatives have demonstrated a protective effect against inflammatory diseases including CRC, with the exception of indoxyl sulfate, a uremic toxin involved in chronic kidney and vascular diseases [100]. Tight epithelial junctions, adequate mucin production and anti-microbial defenses are indispensable for a healthy host physiology; functions all governed via AhR activation by indoles [4,19,90]. Though a more thorough investigation of microbial indoles in early vs. later stages of colon carcinogenesis is warranted given the complex effects on downstream signaling pathways the can both suppress and promote carcinogenesis.

Multiple species of bacteria can produce AhR ligands. *Lactobacillus reuteri* and acidophilus produce 3IAld, which stimulates innate lymphoid cells to release AMP’s to defend against pathogens [57]. *Clostridium Sporogenes* can produce IPA, inhibiting the action of pro-inflammatory signaling molecules, NF-κB and TNF-α, and free radicals [94]. *Clostridium sporogenes*, *Peptostreptococcus russellii*, *Peptostreptococcus anaerobius*, and *Peptostreptococcus stomatis* can produce indoleacrylic acid (IA), shown to attenuate inflammatory response and improve barrier function by mucus production and goblet cell differentiation, potentially via AhR activation [88,101]. *Bacteroides ovatus*, *Bacteroides fragilis*, *Bifidobacterium pseudolongum*, *Clostridium bartlettii*, *Clostridium difficile*, and *Clostridium lituseburense*, among others can produce IA which aids in strengthening intestinal mucosal integrity, possible by activating AhR, and production of IL-22, by mechanisms not well understood [101,102]. Besides indolic ligands, bacteria are also capable of producing non-indole AhR ligands. *Pseudomonas aeruginosa* and mycobacterium tuberculosis are capable of activating AhR by releasing virulence factors including phenazines and phthiocol resulting in pathogen clearance [99,103]. Therefore, bacterial contribution to the pool of AhR ligands should not be overlooked. The mechanistic explanation to the microbial contribution may be limited, but the functional outcome of this contribution is significant, and is evident in the attention the field has received in the recent years.

## 6. Alterations in Microbial Trp Metabolism in Colon Carcinogenesis

As a result of disturbances in host physiology and microbial composition during carcinogenesis, the commensals can take on a pathogenic behavior resulting in an increased inflammation in the host as well as biofilm formation [104,105,106]. About 16–20% of all cancers are the result of pathogenic infections [107]. Patients with Crohn’s disease harbor pathogenic *Escherichia coli* that contribute to a pro-inflammatory state [108]. This pathogenic behavior can directly or indirectly influence changes in host IDO1 expression and cell proliferation [109]. Increases in *Enterococcus faecalis* and *Escherichia coli* enhance the production of intestinal inflammatory signaling molecules, IFN-γ and IL-4, capable of inducing increased expression of IDO1 and altering Trp metabolism [99,110,111]. Another species associated with CRC and IBD, is *Fusobacterium nucleatum*. *F. nucleatum* species isolated from patients with IBD are more invasive and more frequently found in adenomas and adenocarcinomas of the colon [112,113]. Further *F. nucleatum* elicits substantially greater amounts of pro-inflammatory TNF-α gene expression, which can lead to increased IDO1 activation [99,106,114,115]. Both *F. Nucleatum* and *Peptostreptococcus anaerobius* can attach to the cancer cells through cell surface attachment proteins, activating the PI3k-Akt pathway leading to cell proliferation [116,117]. Intriugingly, both of the species can also produce high levels of indoles from Trp metabolism. Therefore, the microbiome is a key contributor to inflammation through altering Trp metabolism [11] which is an important contributing step in colon carcinogenesis.

Changes in microbial indole production is also a characteristic of colon cancer. A reduction in indolic pathway activity was observed in the fecal samples of individuals with CRC. The study demonstrated a lower indole/Trp ratio in CRC when compared to healthy individuals, as well as a higher kyn/Trp ratio [109]. This alteration in the indolic pathway can lead to an increased inflammatory response via TNFα, IL-1β, and IL-6 in colon carcinogenesis, subsequently altering AhR activity [109]. AhR activation is reduced in diseases including IBD, liver disease, metabolic syndrome, autoimmune disease, and cancer, suggesting that alteration in Trp metabolism or indole production my contribute to AhR dysfunction [39,54,55,79]. Fecal samples from individuals with metabolic syndrome, obesity, Type-2 diabetes, and chronic intestinal inflammation also have significantly lower levels of microbial Trp metabolites, potentially contributing to reduced AhR activation. Specifically, individuals with obesity have a significantly higher concentration of Kyn and IDO1 in adipose tissue, which is associated with an imbalance in Th17/Treg cells; suggesting a mechanistic link between altered tryptophan metabolism, obesity and CRC pathogenesis [52,118]. Additionally, administration of microbial indolic metabolites, diindolylmethane or I3C (AhR ligands), reduces tumor formation in *Apc*^Min/+^*AhR*^+/+^ and *Apc*^Min/+^*AhR*^+/-^ mice indicating that specific microbial indoles are protective against carcinogenesis [119]. Treatment with 6-Formylindolo [3,2-b] carbazole (FICZ), a Trp derived AhR ligand, is also effective at restoring intestinal integrity [52]. Intriguingly, *Lactobacillus reuteri* is capable of differentiating T cells into Tregs through AhR activation via indolic compounds, implying additional mechanism to reestablish epithelial barrier function [120]. Similarly, restoration of AhR activation, using *L. reuteri*, reestablished AhR activity in murine model of impaired metabolic syndrome by increasing AhR ligand availability [52]. Collectively, it is implied that a disrupted microbial Trp-indole-AhR pathway significantly contributes to the pathogenesis of CRC. Therefore, identifying the effects of bacterial indoles in the pathogenesis of colon carcinogenesis is important in understanding the impact of the microbiome and potentially developing therapeutics.

## 7. The Contribution of Serotonin (5-HT) in the CRC Microenvironment

Apart from the Kyn and the indolic pathway, 1–2% of Trp can undergo hydroxylation to produce serotonin via the serotonin pathway. As a potent signaling molecule, having 14 classes of receptors, serotonin exhibit diverse effects on the immune function, nervous and endocrine system, psychological processes, blood clotting, metabolic homeostasis, bone metabolism, hematopoiesis, and epigenetic control [121,122]. While serotonin is well recognized as a neurotransmitter of the brain, about 95% of the serotonin is synthesizes and released by the enterochromaffin cells of the intestines, the largest neuroendocrine organ. With majority of serotonin produced in the gut, it plays an important role in maintaining normal gut function, including gut motility, absorption, and vasodilation [121,123]. Therefore, the role of serotonin in a healthy physiology is indispensable.

While serotonin is essential in healthy physiology, studies have demonstrated both protective and detrimental role of serotonin in colon carcinogenesis [124], where a detrimental role is taken over in response to an altered or impaired serotonin activity [125]. Serotonin exhibits both pro- and anti-inflammatory signaling activity in the intestines via activation of 5-HT 7 and 5-HT4 receptors, respectively [121]. Studies have also confirmed the role of serotonin in tumor growth, cell invasion, angiogenesis, and metastasis in CRC [126,127,128]. A β-catenin mutation (CtnnB1) occurring during early phases of stem cell differentiation into enteroendocrine cells, has shown to induce tumors expressing serotonin [127,129]. Activation of serotonin receptors, 5-HTR1B and 5-HTR2B stimulate tumor angiogenesis and cell proliferation [130,131]. Receptor 5-HT1D, by activating Axin1/β-catenin/MMP-7 pathway, also promote tumor cell invasion. The same study demonstrated the inhibition of tumor metastasis by administering 5-HT1D antagonist, GR127935, in a mouse model [125]. Early interventions with Mirtazapine, inhibitor of serotonin 2 C receptor (HTR2C), has shown to reduce tumor growth and prolong survival rate in tumor-bearing mice. This reduction in tumor growth was due to activation of immune response and restoration of serotonin levels [132]. The above evidence supports the dual role of serotonin in tumor prevention and proliferation, due in part to altered serotonin pathway, mutations, and inflammation.

The microbiota plays a significant role in regulating serotonin levels, synthesized and released by the enterochromaffin cells of the intestines [121]. Short chain fatty acids produced by the microbiota in response to saccharides in the diet, stimulate the enterochromaffin cells to produce and release serotonin [133]. Spore-forming bacterial have shown to produce metabolites that promote serotonin synthesis, mechanistically unknown. This microbiota-induced serotonin production can affect gut motility and intestinal homeostasis, suggesting a therapeutic intervention [134]. Several microbial species including *Corynebacterium* spp., *Escherichia coli*, and *Streptococcus* spp., can also produce serotonin [134]. Serotonin-like molecules are also secreted by *Rhodospirillum rubrum*, *Bacillus cereus*, *Enterococcus faecalis*, and *Staphylococcus aureus* [135]. Collectively, the intestinal and microbial serotonin production is involved in several functions including gut motility, platelet function, and immune response under normal physiological conditions [134,136]. However, serotonin uptake and production can become dysregulated in pathogenesis of several diseases including colon cancer [124,134].

Serotonin is essential in protecting against cellular damage and disease development. Serotonin activity protects the intestines from DNA damage [21,127] and induces signaling (HTr4 receptor; effect-gut motility) that aids in reducing the development of colitis and controlling the development of early CRC [137]. Under the conditions of immune homeostasis and a diverse microbiota, serotonin exerts a protective role [133,134,138]. A disruption in the balance of the immune system and the microbial composition allows serotonin to instead promote cell proliferation and angiogenesis [124,127,128]. Specifically decreased serotonin activity is associated with DNA damage, apoptosis, and constricted arterioles in CRC leading to a promotion of carcinogenesis [127,139,140]. A study comparing the metabolomic profile of cancer and normal tissues demonstrated that Trp metabolites; serotonin and 5-Hydroxy-3-indoleacetic acid [5-HIAA]) were only detected in normal tissue and absent in tumor tissue, indicating that the tumor microenvironment was actively metabolizing serotonin [141]. Collectively, the following evidence demonstrates the dual role of serotonin in promoting intestinal homeostasis and accelerating carcinogenesis.

## 8. The Contribution of Tryptamine in CRC Microenvironment

Decarboxylation of Trp leads to production of tryptamine, a monoamine alkaloid with an indole ring in its structure [142]. Tryptamine is mainly found in the central nervous system, but can also be produced by bacterial, fungi, and plants [143]. Under normal physiology, tryptamine acts as a neurotransmitter, vasoconstrictor and vasodilator, anti-oxidant and antibacterial agent [144]. Monoamine oxidase (MAO)-mediated tryptamine metabolism also provides an endogenous source of AhR ligand, thereby inducing CYP1A1 expression [143]. Tryptamine can also induce the enterochromaffin cells to release serotonin, fluctuations of which have shown to play a significant role in pathology of inflammatory disease, a risk factor for colon carcinogenesis [145].

Tryptamine is also produced by the microbiome, which aids in colonic fluid secretion via activation of serotonin receptor HT-4, subsequently affecting colonic transit in a mouse model. Mechanism by which tryptamine increases colonic fluid secretion remains unclear, due in part to the structural similarities of tryptamine to serotonin and ligand-binding ability of tryptamine to serotonin receptors [23]. Given that ~50% of all patients undergoing chemotherapy for colon cancer experience chemotherapy-induce diarrhea, understanding the impact of microbial tryptamine production would be important to abrogate this serious side effect. Specific bacteria have been identified that can perform decarboxylation of Trp into tryptamine, a common practice in plants, but rare among the bacterial species [146]. These bacteria include *Xenorhabdus nematophilus* and *Bacillus atrophaeus*. *Lactobacillus bulgaricus* can also excrete tryptamine [147,148]. Therefore, engineered tryptamine producing bacteria may be a potential therapeutic intervention to increase serotonin levels and AhR ligand availability, but may have negative effects in the context of colon carcinogenesis and treatment.

## 9. Tryptophan Metabolism and Bacterial Indoles as Therapeutic Targets

Both host and bacterial-Trp metabolites play a significant role in colon cancer initiation and development. Increased expression of Kyn and IDO1, and reduced production of indoles have been detected in variety of cancers including CRC [149,150]. Increased activity of IDO1 in conjunction with Trp depletion, leads to cell cycle arrest in activated T lymphocytes. This increased IDO1 activity further leads to apoptotic T-cell death [151,152,153]. Together, evidence indicates that increased IDO1 activity and Trp depletion promotes immunosuppression in the tumor microenvironment, making IDO1 a promising target for therapy in combination with traditional chemotherapeutics [154,155]. Additionally, evidence demonstrates that engineered microbes targeting the IDO1 pathway have therapeutic benefits in animal models of colon cancer [156,157]. Therefore, small molecule compounds with immunoregulatory effects, including both chemical and microbial IDO1 inhibitors, seem hopeful as therapeutics. Yet, it is unknown whether microbial indole production is protective or deleterious in the context of increased IDO1 or AhR expression in colon cancer. We present the following therapeutic modalities to be considered for study as interventions among individuals with CRC and high IDO1 or AhR expression, and discuss the current understanding and opportunities for future research around Trp metabolism and indole production for therapeutic intervention in CRC.

### 9.1. Therapeutic Targeting of Indoleamine 2,3-Dioxygenase (IDO1)

The overexpression of IDO1 is not merely a consequence of tumorigenesis but is seen as an early hallmark of inflammatory bowel disease and colitis indicating that it is in part driving tumorigenesis [158,159,160,161]. Loss of tumor suppressor gene bridging integrator 1 (BIN1) and overexpression of pro-inflammatory enzyme, COX-2 are also associated with an increased IDO1 expression, along with signals from pro-inflammatory cytokines, especially IFNγ [162]. IDO1 overexpression is also observed in colon cancer cell lines (HCT-116 and HT-29) in absence of an inflammatory environment, suggesting that genetic mutations or epigenetic modification of IDO1 could also be driving colon carcinogenesis [32,163,164]. Evidence from several seminal studies have clarified the carcinogenic role of IDO1, the blocking of which inhibits tumorigenesis and proliferation [164,165]. Not only has IDO1 inhibition been protective against chronic inflammation, it has been successful at altering microbial composition, resulting in increased indolic metabolites and reestablishing gut permeability and intestinal homeostasis [13,95,161].

Based on the role of IDO1 in cancer immunity and development, several small molecule drugs have been developed to inhibit IDO1 for many types of cancer including CRC [166]. Recently Liu et al. showed that I-L-MT, an IDO1 inhibitor, decreased expression of cell cycle gene CDC20 resulting in G2/M arrest in HCT-116 and HT-29 cells. I-L-MT also induced mitochondrial injury and apoptosis in cancer cells [167]. Other effector IDO1 inhibitors in clinical trials include Indoximod (trial completed), Epacadostat (trial completed), Navoximod (trial completed), BMS-986205 (Recruiting), and PF-06840003 (trial not active) [166]. Most recently, Shen et al., constructed a liposome delivery system containing oxaliplatin and NLG919 (IDO1 inhibitor), to test the efficacy of this treatment in a mouse model of CRC, proposing an effective therapeutic approach for reversing the immunosuppressed tumor environment [168]. Ipilimumab, a CTLA-4 blocking antibody can also inhibit IDO1 expression [168]. These drugs were capable of not only inducing immunogenic cell death, but also reducing the conversion of Trp into Kyn through IDO1 inhibition. In addition to the clinical trials for IDO1 inhibitors, there have been clinical trials for vaccination using IDO1 peptides [169]. In lieu of the above treatment trials, there is an ample body of literature to support further investigate the oncoenzyme, IDO1 and IDO1 blockade in host cells. This is especially important for colon cancer therapy which appears to be more refractory to immunotherapy as compared to other types of cancer and thus would benefit from drugs that enhance the T cell response.

In addition to studying IDO1 blockade using traditional drug target methods, bacterial species are being used or engineered to block IDO1 expression. Specifically, *Lactobacillus johnsonii* decreased IDO1 activity (47%) in HT-29 intestinal epithelial cells by producing hydrogen peroxide (H2O2) which can inhibit IDO1 activity [157]. Additionally, *Bifidobacterium Infantis* decreased IDO1 activity and Kyn-Trp ratio, which reduced inflammation in murine model of depression. In contrast, *B. Infantis* can enhance levels of kynurenic acid, a protective metabolite, which inhibits colon cancer in a murine model of inflammation [7,170]. The ability of *Lactobacillus reuteri* to reduce pro-inflammatory cytokines is an important step in reducing IDO1 expression, which is tightly regulated by INF-y [171]. *L. reuteri*, which can reduce NF-kB activity, inhibit cell proliferation, and promote apoptosis of colon cancer cells [172] has been used in clinical trials (NCT03501082) to improve gut microbiome function. Additionally, Phan et al. investigated the efficacy of *Salmonella typhimurium* delivered shRNA targeting IDO1 inhibition (shIDO1-ST) which resulted in reduced tumor growth in two mouse models of CRC [173]. Bacterial species capable of producing short chain fatty acids, especially butyrate and propionate, can inhibit INF-y-induced IDO1 expression via downregulation of STAT-1 and reduced HDAC inhibition activity [75]. Interestingly, Zelante et al. demonstrated that in the absence of IDO1 expression (IDO1 knock out), *Lactobacilli* were increased leading to an increased production of AhR-activating indolic compounds including indole-3-lactic acid and indole-3-aldehyde, suggesting that IDO1 expression also controls microbial composition and indole production. Further, the AhR activation via indolic compounds increased transcription of IL-22 which stimulated the epithelial immune response, and reduced pathogen abundance via recruitment of neutrophils and macrophages [13,174,175]. Therefore, the utility of *L. reuteri*, *B. Infantis*, and *L. johnsonii*, as well as engineered bacteria, as therapeutics in CRC is of great interest. The above evidence suggests a significant role of the microbiome in manipulating IDO1 expression and visa versa. Therefore, not only are drug targets of IDO1 promising therapeutics for CRC, but also are several microbial species.

### 9.2. Therapeutic Targeting of Aryl Hydrocarbon Receptor (AhR)

AhR plays a significant role in intestinal inflammation and colon carcinogenesis. Several studies have proposed AhR as a target for anticancer therapy [26,176]. Under inflammatory conditions, including CRC, there can be either partial or complete loss of AhR, whereas xenobiotic induced AhR activation has shown to reduce colitis in experimental models [177]. Therapeutically targeting intestinal AhR activation is problematic due to the multiple AhR ligands, and plethora of downstream AhR target genes that have disparate effects [13,177,178]. Despite these complexities, the role of AhR in therapeutics for tumor suppression is under intense investigation. Several AhR ligands were recently in clinical trials for renal and breast cancer, autoimmune diseases, and multiple sclerosis [179,180,181,182]. Recently, however, a trial using an AhR inhibitory drug (BAY2416964) was initiated for treatment of advanced colon cancer (NCT04069026). Results from this trial will be important to understand the effect of AhR modulation in colon cancer progression.

Bacterial indoles, as a class of AhR ligands, are also being studied extensively in colon carcinogenesis. Depending on the concentration and structure of the indole compound, they can act as either agonists or antagonists of AhR [61]. Bacterial indoles, indole-3-acetate, tryptamine, IA, and indole-3-pyruvic acid can act as AhR agonists, while indole and IPA are AhR antagonists [177,183]. Both dietary and bacterial indoles exhibit promising therapeutic targets for abrogating carcinogenesis. [17,95,184]. However, more research is required as indoles demonstrate concentration dependent and tissue specific outcomes [83,185].

### 9.3. Therapeutic Targeting of Oncogene, c-Myc, to Control Trp Metabolism

It is well established that overexpression of oncogene, c-Myc, which is evident in about 70% of all colon cancer cases [186,187,188], is an important metabolic reprogramming in colon cancer initiation and progression [189,190]. The oncogene c-Myc dysregulates the vital cellular functions including survival, apoptosis, differentiation and proliferation, and promotes angiogenesis, metastasis, and immune evasion [188,190,191,192,193]. Most important, overexpression of c-Myc increases Trp uptake and conversion to Kyn stimulating proliferation of colon cancer, indicating Myc as a promising therapy in colon cancer to reduce Trp uptake [43].

Several strategies have been deployed to target c-Myc directly or indirectly including inhibiting c-Myc transcription, translation, activity, stability, gene targets, and interactions via different pathways including PI3K-mTOR to control CRC progression [194]. Wiegering et al. demonstrated silvestrol inhibited c-Myc expression reducing proliferation of colon cancer cells in vivo [195]. Unfortunately, targeting c-Myc has not been as successful in the trials due to several concerns. These reasons include the essential role of c-Myc in tissue homeostasis, and presence of partially redundant transcription factors (MYC, MYCN, MYCL), which would need to be targeted together for efficient therapeutics [196]. Despite these challenges, c-Myc continues to be an interesting and promising target for colon cancer, especially due to its upregulation of Trp metabolism, the blocking of which has been successful at reducing tumor proliferation.

### 9.4. Trp Metabolizing Bacteria as a Therapeutic Target

Apart from drug compounds, engineered bacterial species may be another avenue for therapeutics modulation of Trp metabolism in colon cancer. Several bacterial species carry the gene for tryptophanase (TnaA), an enzyme that degrades Trp to indoles, including *Escherichia coli*, *Proteus vulgaris*, *Bacillus alvei*, *Porphyromonas gingivalis*, *Haemophilus infuenzae*, *Proteus rettgeri*, and *Aeromonas liquefaciens* [197]. Another gene cluster was identified by Wlodarska et al. with the capacity to metabolize phenylalanine to indole compounds including IPA and IA. The gene cluster, *fldAIBC*, was identified in *Peptostreptococcus russellii*, *Peptostreptococcus anaerobius* and *Peptostreptococcus stomatis*, which demonstrated the ability to utilize mucins, and produce indoles that inhibited inflammatory response [88]. Similarly, *Clostridium sprogenes* can metabolize Trp into IPA, protecting against gut permeability and inflammation [198]. Several other Trp-metabolizing phyla including Actinobacteria, Firmicutes, Bacteroidetes, Fusobacteria, Proteobacteria, and genera including *Burkholderia*, *Streptomyces*, and *Pseudomonas* have been recognized to possess this function [199]. Kaur et al. suggested Fusobacteria as one of the highest indole producing phyla with about 72% of the Fusobacteria strains containing genes associated with indolic pathways [199]. Interestingly, *Fusobacterium nucleatum* is one of the most abundant bacteria in colon adenomas and in colon adenocarcinomas and is associated with colon cancer promotion [200]. Additionally, biofilm formation is a key hallmark of adenomas and colon carcinomas. It is intriguing that *Fusobacterium nucleatum* is one of the strongest producers of indole, which is also known to promote biofilm formation. Yet, bacterial indole production is a strong anti-inflammatory factor in the host. Thus, an outstanding question in the field of colon cancer is whether *F. nucleatum*, or similar indole producing bacteria, is contributing to biofilm formation through indole production, and whether indole is acting in a protective manner or not in the context of colon cancer?

### 9.5. Dietary Indoles as Therapeutic Targets

In addition to indoles and Trp metabolites, AhR activation can be achieved by dietary compounds, collectively called brassica metabolites. These include 3-3-diindolylmethane (DIM), Indole-3-carbinol (I3C), Indole-3-acetonitrile (I3ACN) and, Indole[3,2b] carbazole (ICZ), which can be found in cruciferous vegetables including cabbage, broccoli, kale, collards, turnip greens, mustard, radish, rapeseed, kohl rabi and brussel sprouts.

Cruciferous vegetables are widely studied in cancer prevention, including colon cancer [3,201,202]. These vegetables are a good source of glucosinolate, predominantly, glucobrassicin, which can be broken down into indolic compounds. The enzymatic cleavage of glucobrassicin by myrosinases produce indolic compounds including I3C and I3ACN, which can be further reduced to DIM and ICZ. Collectively, these indolic compounds can activate AhR, and have demonstrated a reduced cancer risk [13,80,203,204,205,206,207]. However, it is unclear how liver metabolism of dietary indoles contributes to negative effects through formation of indoxyl sulfate (aka. indican) that can cause oxidative stress.

Specifically, I3C is emerging as an anti-carcinogenic molecule, exhibiting anti-proliferative effects in several colon cancer cell line including HT29, Colo320, Caco-2, HCT-116, and WS480 [3,208,209]. The protective role of I3C is not only seen in cancer but also evident as a preventive factor in inflammatory diseases [210,211,212]. Mechanistically, I3C reduces inflammation and carcinogenesis through suppression of NF-κB signaling pathway, and induces cell cycle arrest, and apoptosis [183,213,214,215,216]. Additionally, I3C has demonstrated beneficial effects on the microbiome by promoting the production of antimicrobial factors that inhibit pathogenic biofilm formation. However, higher doses of I3C increased colonic lesions and tumor initiation due to overactivation of AhR in a murine model of hepatocarcinogenesis [13,217,218]. Therefore, the anti-cancer properties of dietary indole compounds may overcome the pathogenic indole production and reduce colon carcinogenesis in a dose and context dependent manner; however, much more research will be required to precisely deteremine the dietary vs. bacterial contribution to carcinogenesis.

## 10. Conclusions

Trp metabolism and indoles, both dietary and bacterial, are key factors in maintaining normal host physiology, including immune homeostasis, gut barrier function and microbial composition. However, under inflammatory conductions such as colon cancer, a series of alterations that affect host and microbial Trp metabolism occur. One key alteration includes overexpression of IDO1 leading to an increase production of Kyn. Kyn can constitutively activate AhR to enhance Treg activity, promoting immune escape. In contrast, Trp metabolites, including indoles, can activate intestinal AhR to suppress colonic stem and progenitor cell growth and reduce tumorigenesis in vivo; potentially ameliorating cancer stem cell proliferation. Furthermore, the changes in the gut microbiome during colon carcinogenesis also result in a paucity of indoles from Trp metabolizing bacteria. Subsequently, the reduced availability of protective indolic metabolites disrupt the symbiosis among microbial communities leading to an enhanced inflammation response. This heightened inflammatory condition further increases IDO1 activity driving a vicious cycle of tumor escape, proliferation, and metastasis. As such, increasing dietary indoles or promoting the growth of Trp metabolizing bacteria (e.g., Lactobacillus) to produce indoles may act to reduce the availability of Trp to cancer cells.

Harnessing the Trp-indole pathways in bacteria and the host hold the potential to attack the tumor cells through several different facets; Trp starvation, IDO1 inhibition, and indolic AhR activation. Further investigation is required to identify specific ligands that control AhR activation and mechanisms that reduce IDO1 expression or activity. Both dietary and microbial indoles hold promise as therapeutic adjuvants to target these pathways to inhibit colon carcinogenesis.

## Figures and Tables

**Figure 1 nutrients-13-01189-f001:**
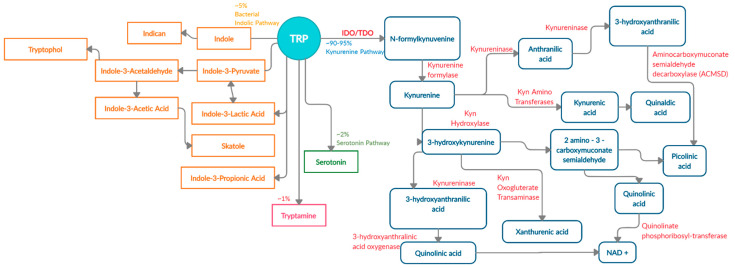
Schematic representation of enzymes and metabolites in the Trp-Kyn pathway.

**Figure 2 nutrients-13-01189-f002:**
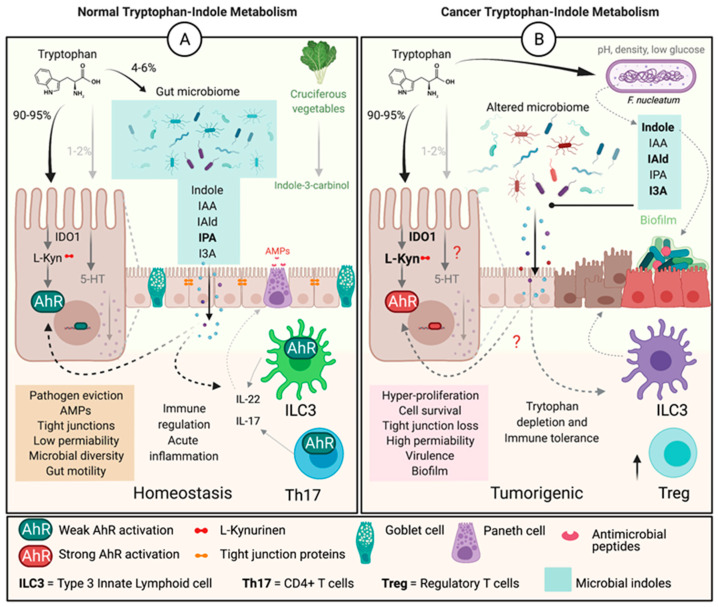
Tryptophan-indole metabolism in normal vs. colon carcinogenesis. (**A**) Under normal physiological conditions, Trp is mainly metabolised by hepatic or extrahepatic cells by IDO1 into L-Kyn, with the rest being metabolised by bacteria or converted to 5-hydroxytryptamine or seratonin. The bacteria convert Trp to indoles, and together with dietary indoles, IDO1 and L-Kyn are ligands for AhR. Indoles also regulate the production of cytokines in by ILC3 and Th17 cells, which result in production of AMPs and improved barrier function to regulate inflammation and maintain homeostasis. (**B**) In a tumorigenic environment IDO1 is overexpressed and Trp is depleted driving L-Kyn levels higher and constiuatively activating AhR. High L-Kyn leads to potentially stronger AhR activation and overproduction of IL-22/IL-17 and hyperproliferation, along with activation of Treg cells creating an immunotolerant environment permissive for tumor growth. It is unclear (?) how microbial indolic metabolites control epithelial or immune cell metabolism in the context of colon cancer.

**Figure 3 nutrients-13-01189-f003:**
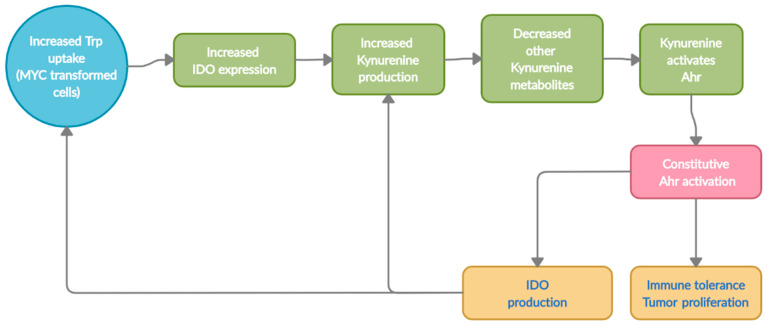
Sequential representation of Trp metabolism in Tumors (host cells).

**Table 1 nutrients-13-01189-t001:** Classes of Ahr ligands.

Class of Molecule	Metabolites	Genes Activated	Effects (Ligand Dependent)
Tryptophan Metabolites	Kynurenine Kynurenic Acid Xanthurenic Acid Anthralinic Acid Quinolinic Acid Pinolinic Acid Cinnabarinic Acid 3-hydroxykynurenine 3-hydroxyanthralinic acid	CYP1A1/CYP1B1 AHRR IL-6 VEGFA PTGS2 IL-22 IDO1 TDO	Neurotoxic Effects B-cell & T-cell differentiation Development of intraepithelial lymphocytes Immune tolerance Enhanced epithelial barrier function Anti-inflammatory Intestinal homeostasis Gut motility Microbial composition Antimicrobial affects Mucosal barrier function Serotonin modulation
Dioxins	Polycyclic Aromatic hydrocarbons Benzatheracues Halogenated aromatic hydrocarbons		
Dietary compounds	3-3-diindolemethane (DIM) Indole-3-carbinol Indole-3-acetonitrile Indole [3,2-b] carbazole 2-(indole-3-ylmethyl)-3,3-diindolymethane Herbs- Ginseng, gingko biloba, licorice		
Microbial metabolites	Indole 3-hydroxyindole Indolealdehyde Tryptamine Indole acetic acid Tryptanthrin 3-methyl indole Indirubin Indigo Indole sulphate Malassezin		
Photo-oxidative Trp metabolite	6-formylindole [3,2-b] Carbazole (FICZ)		

## Data Availability

Not applicable.

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
