# Peer review of "Targeting Dietary and Microbial Tryptophan-Indole Metabolism as Therapeutic Approaches to Colon Cancer"

_nutrients, 2021, doi:10.3390/nu13041189_

Round 1

Reviewer 1 Report

Concerns:

  1. Lane 214 or 361- Since it has been demonstrated that Trp metabolism is activated in human obesity and  likely results  from  the  inflammatory  response  linked  to this  pathology (Am J Physiol Regul Integr Comp Physiol. 2012; 303(2):R135–43), I recommend to include the part regarding possible connection of CRC to obesity in the view of Trp metabolism
  2. In my opinion -and it is rather a recommendation- Chapter 9 can benefit from some shortening, perhaps by combining 9.1 and 9.2 parts.

Author Response

  1.  Lane 214 or 361- Since it has been demonstrated that Trp metabolism is activated in human obesity and  likely results  from  the  inflammatory  response  linked  to this  pathology (Am J Physiol Regul Integr Comp Physiol. 2012; 303(2):R135–43), I recommend to include the part regarding possible connection of CRC to obesity in the view of Trp metabolism

Author response: Thank you for this important insight. We have added this citation and additional text “Specifically, individuals with obesity have a significantly higher concentration of Kyn and IDO1 in adipose tissue, which is associated with an imbalance in Th17/Treg cells; suggesting a mechanistic link between altered tryptophan metabolism, obesity and CRC pathogenesis”

  1. In my opinion -and it is rather a recommendation- Chapter 9 can benefit from some shortening, perhaps by combining 9.1 and 9.2 parts

Author response: We appreciate your recommendation, however after deliberation, we decided it was more meaningful to keep 9.1 and 9.2 separate.

Reviewer 2 Report

Review by Wyatt and Greathouse  onTargeting dietary and microbial tryptophan-indole metabolism as therapeutic approaches to colon cancer is very well written covering broad area of information. Authors cover different metabolic pathways of Tryptophan in colon carcinogenesis and reviewing the influence of microbiome during colon carcinogenesis.

Author Response

Author response: We appreciate the supportive comments of this reviewer but did not note any specific recommendations for changes.

Reviewer 3 Report

Authors in presented article entitled: "Targeting dietary and microbial tryptophan-indole metabolism as therapeutic approaches to colon cancer. " reviewed scientific literature on tryptophan metabolism with focus on role of metabolites of tryptophan in pathophysiology of inflammation and pathogenesis of colon cancer. It is a well-written review evaluating many interesting aspects of tryptophan metabolism.

Broad comments

Authors reviewed tryptophan metabolism in more practical way with possible therapeutic approches and smoothly included both basic science studies and clinical trials in this review. Additionally, by reviewing separately metabolic pathways authors provided organized article, easy to read. Despite these positive aspects there are a few editing errors that should be corrected.

Specific comments

  1. Figure 1 shows complex metabolism of tryptophan and it is great that authors decided to include this figure. However, there is indole-3-acetic acid written 3 times, what is misleading. Additonally authors did not include indole-3-propionic acid in this figure. It should be corrected.
  2. On table 1 there is "free radical formation" at the bottom of the table. Should it be there? In first collumn there are classes of molecules and it looks more as a function.
  3. Authors should check manuscript and correct abbreviations:
    - CRC was written many times, I suspect it means colorecal cancer. However, authors did not explain this abbreviation in text.
    - line 290 - indole and there IPA in bracket (IPA can be abbrev for indole-3-propionic acid)
    - line 316 authors decided to write IPA after indole-3-propionic. However, name of this particle was used at least twice before this line in the manuscript. Abbreviations should be after molecules was first introduced to the text.
  4. Spelling should be corrected in lines:
    -line 336 - Crohn's disease not Chron's disease
    -line 483 and 621 authors once wrote about HTC-116 and once about HCT-116 cell line.

Author Response

  1. Figure 1 shows complex metabolism of tryptophan and it is great that authors decided to include this figure. However, there is indole-3-acetic acid written 3 times, what is misleading. Additonally authors did not include indole-3-propionic acid in this figure. It should be corrected.

Author response: Thank you for bringing this error to our attention. It has been corrected.

  1. On table 1 there is "free radical formation" at the bottom of the table. Should it be there? In first collumn there are classes of molecules and it looks more as a function.

Author response: Thank you for allowing us to make this correction. The table has been updated.

  1. Authors should check manuscript and correct abbreviations:
    - CRC was written many times, I suspect it means colorecal cancer. However, authors did not explain this abbreviation in text.

Author response: Thank you for pointing out this error. It has been corrected.

- line 290 - indole and there IPA in bracket (IPA can be abbrev for indole-3-propionic acid)

Author response: Thank you, we have made this correction.

- line 316 authors decided to write IPA after indole-3-propionic. However, name of this particle was used at least twice before this line in the manuscript. Abbreviations should be after molecules was first introduced to the text.

Author response: Thank you for bringing this error to our attention. It has been corrected.

  1. Spelling should be corrected in lines:
    -line 336 - Crohn's disease not Chron's disease
    -line 483 and 621 authors once wrote about HTC-116 and once about HCT-116 cell line.

Author response: Thank you for bringing this error to our attention. It has been corrected.